

# What caused the interdecadal shift of the ENSO impact on dust mass concentration over northwestern South Asia?

Lamei Shi [1, 2], Jiahua Zhang [1, 2], Fengmei Yao [2], Da Zhang [1, 2], Jingwen Wang [1, 2], Xianglei Meng[2], Yuqin Liu[3]

[1] Key Laboratory of Digital Earth Science, Aerospace Information Research Institute, Chinese Academy of Sciences, Beijing 100094, China

[2] College of Earth and Planetary Sciences, University of Chinese Academy of Sciences, Beijing 101407, China

[3] Key Laboratory of Urban Environment and Health, Institute of Urban Environment, Chinese Academy of Sciences, Xiamen 361021, China

*Correspondence to:* Jiahua Zhang (zhangjh@radi.ac.cn)

**Abstract.** The change of large-scale circulation, especially El Niño-Southern Oscillation (ENSO), play an important role in the interdecadal variability of dust activities over the dust source and downwind regions. However, the detailed factors that lead to the interdecadal variability of the ENSO impact on dust activities over the northwestern South Asia remain less clear, although previous studies have discussed the response of the interannual dust activities over the northwestern South Asia to the ENSO circle. Based on the linear regression model and MERRA-2 atmospheric aerosol reanalysis data, this study investigated the interdecadal variability of the ENSO impact on dust activities as well as the associated possible atmospheric drivers under two different warming phases over the northwestern South Asia. Results indicated that the relationship between ENSO and surface dust mass concentration (DUSMASS) experienced an obvious shift from the accelerated global warming period (1982–1996) to the warming hiatus period (2000–2014). The change of Atlantic SSTA pattern weakened the impact of ENSO on dust activities over the northwestern South Asia, while that of Indian Ocean SSTA pattern and PDO tended to strengthen ENSO's effect. Both the Atlantic and Indian Ocean SSTA patterns were modulated by the duration of ENSO events (i.e., continuing and emerging ENSO). The Eurasian continent and Indian Ocean thermal contrast was less likely to cause the shift of ENSO–DUSMASS relationship. This study provides new sights to numerical simulation involving the influence of atmospheric teleconnections on the variability of dust activities and their influence mechanisms.

**Keywords:** Surface dust mass concentration; ENSO–DUSMASS relationship; interdecadal change; large-scale atmospheric circulation; northwestern South Asia



## 1 Introduction

Dust aerosol is attracting an increasing concern due to its adverse impacts on human health
(Bozlaker et al., 2013; Chen et al., 2004; Erel et al., 2006; Kaiser and Granmar, 2005; Poulsen et al.,
1995; Sanchez de la Campa et al., 2013; Schulz et al., 2012) and environmental problems (Avila, 1998;
Behrooz et al., 2017; Razakov and Kosnazarov, 1996). The Northwest Indian subcontinent, which is the
most arid and semiarid area of South Asia, suffers heavy and frequent dust storms in summer due to
extremely dry climate and strong winds (Jin and Wang, 2018). Those dust storms can travel long-distance
to North India and the Arabian Sea, degrading air quality (Mahowald et al., 2010) and modifying ocean
biogeochemistry processes (Richon et al., 2018; Singh et al., 2008). Particularly, dust aerosols can change
local radiation budget, circulations, and Indian summer monsoon rainfall through absorption and
scattering of solar radiation. The mineral dust over the northwestern South Asia is closely associated with
the long-term variation of global climate (Banerjee et al., 2019; Bollasina et al., 2011; Jin et al., 2018).
To better understand such feedback, and so that to give early warning to reduce disasters and losses
caused by dust events, it is important to find out the controlling factors of the surface dust mass
concentration (DUSMASS) and its long-term variation.
ENSO, as a periodic fluctuation in sea surface temperature (SST) and the air pressure across the
equatorial Pacific Ocean, is as the primary large-scale driver of dust loading over the global dust source
region (Trenberth et al., 2014). The Niño index significantly impacts the dust activity over the South Asia
either indirectly through dust transport from Southwest Asia and/or directly through precipitation effect
on dust emission (Banerjee et al., 2019; Bollasina et al., 2011; Jin et al., 2018).
An interdecadal climate regime shift was observed in the large-scale boreal winter circulation
pattern over the North Pacific in the mid-1970s (Graham, 1994; Nitta and Yamada, 1989; Trenberth and
Hurrell, 1994). Another remarkable climate change was observed in the early 21st century, i.e., an
accelerated global warming prevailed before late 1990s and a warming hiatus dominated after that
(Easterling and Wehner, 2009; Fyfe et al., 2011, 2013). After 2013, the global warming came to an end
due to a persistent warm condition over the equatorial Pacific between Mar. 2014 and May 2016 (Hu and
Fedorov, 2017). Concurrent with the Pacific climate shift, the large-scale circulation pattern and their
atmospheric teleconnection also exhibit interdecadal change. Statistically 1980–1999 was characterized
by a predominance of eastern equatorial Pacific (EP) and continuing (CT) El Niño event (McPhaden et


al., 2011; Yang and Huang, 2021), while the central equatorial Pacific (CP) and emerging (EM) El Niño
became more frequent since the beginning of the 21st century (Yang and Huang, 2021). The tropical
Pacific and Indian Ocean SST showed a rapid shift from a cold to warm state around the late 1970s. This
climate regime shift altered the links of ENSO with Indian summer monsoon rainfall (ISMR) (Kumar et
al., 1999). The changes of the teleconnection relationship at the turning point of mid-1970s have been
well documented, while that occurred around the early 21st century, particularly the effect of ENSO on
DUSMASS over the northwestern South Asia, is insufficiently analyzed.

The impact of ENSO on DUSMASS over the northwestern South Asia experiences interdecadal

shift. In the context of global warming (Deser et al., 2017; Kosaka and Xie, 2016), the relationship
between El Niño and monsoon experiences interdecadal change. The correlation between El Niño and
rainfall over India turned to be insignificant from the late 1970s, simultaneously, the relationship between
ENSO and monsoon also weakened around this turning point (Kumar et al., 1999). Two influence
mechanisms are proposed to explain this weakened ENSO–monsoon relationship. One is the varied
location of Walker circulation that adjusts the monsoon rainfall over Indian region, the other is the
temperature change over Eurasia that modulates the land-sea thermal gradient. Besides, the impact of
Atlantic Ocean pattern on the monsoon circulation over the Indian Ocean became stronger since late
1970s as the influence of the tropical Pacific has reduced (Kucharski et al., 2007; Sabeerali et al., 2019;
Srivastava et al., 2019). This in-turn impacts the circulation responsible for dust uplift and transport.
Several studies show that the dust activities over the northwest Indian Ocean were also affected by the
Indian Ocean dipole, which modulated the ENSO-related moisture (Banerjee and Kumar, 2016).
However, Agrawal et al. (2017) indicated that whenever the relationship of ISMR with IOD is weaker
(stronger), it becomes stronger (weaker) with the ENSO index. The two types of ENSO are also found
to differ in their links with ISMR, i.e., the central equatorial Pacific El Niño (CP El Niño) shows higher
correlation with Indian droughts than the eastern equatorial Pacific El Niño (EP El Niño) (Kumar et al.,
2006). In addition, the Pacific Decadal Oscillation (PDO) is reported to amplify the effect of ENSO when
it is in phase with ENSO (Roy et al., 2003).

In short, ENSO primarily influences the DUSMASS over the northwestern South Asia by impacting

the winter precipitation and the subsequent soil moisture, but the effects of ENSO on ISMR experienced
remarkable interdecadal change and many factors may cause this transition. Till now, however, the
interdecadal variability in the links of DUSMASS over the northwestern South Asia with ENSO has not





been investigated in detail compared to the North African and West Asian counterpart (Yu et al., 2015).
In addition, though many factors were proved to influence the short-term (e.g., interannual scale)
variation of the relationship between ENSO and dust activities over South Asia, their effects on the long-
term (e.g., interdecadal scale) change were still unclear. Cai et al. (2014) pointed out that global warming
will have a significant impact on ENSO. The extreme El Niño events will become more frequent under
the changes of atmospheric convention in the next half of the 21$^{st}$ century. Thus, understanding the
physical mechanism of the shift of the ENSO–DUSMASS relationship is of profound implications for
the forecast of dust trend in the future climate change scenario.

Many of the past researches on dust events were based on the aerosol optical depth (AOD) data

provided either by observational data at meteorological stations, which are sparsely distributed in the key
dust sources, or by satellite remote sensing with limited coverage and bias caused by cloud contamination
and uncertainty in retrieval algorithms, respectively (Zhang and Reid, 2009). Due to these limitations,
some of conclusions on the effect of ENSO events on dust activities remained contradictory. The current
study used the Modern-Era Retrospective Analysis for Research and Applications, version 2 (MERRA-
2) (Gelaro et al., 2017) atmospheric aerosol reanalysis data. The MERRA-2 can provide high-quality
dust aerosol variables related to emission, transport, and deposition process benefiting from the
integrated multiple satellite systems and ground-based AERONET (Randles et al., 2017). Another
advantage of MERRA-2 is its continuity in both temporal and spatial coverage (Gelaro et al., 2017).
Those are of great significance to explore the interactions between DUSMASS and large-scale
atmospheric circulation.

This study aims to investigate the large-scale atmospheric factors that contribute to the interdecadal

variability of the ENSO impact on DUSMASS over the northwestern South Asia. The paper is organized
as follows. Section 2 describes the datasets and methods. Section 3 presents factors that influence the
interdecadal change of the relationship between DUSMASS and wintertime Niño-3 index. Section 4
discusses the deficiency and prospect of this study. The conclusions are given in Sect. 5.
**2 Data and methods**
**2.1 Study area**

The main dust source over South Asia is a large arid region in the northwestern part of the Indian





subcontinent, which stretches from India to Pakistan. Most of the dust aerosols over this region come
from the Thar desert. The southeastern part of the Thar desert lies between the Aravalli Hills. The desert
extends to the Punjab Plain in the north and northeast, to the alluvial plains of the Indus River in the west
and northwest, and to the Great Rann of Kutch along the western coast. The desert presents an undulating
surface, with high and low sand dunes separated by sandy plains and low barren hills. The soils are
mainly consisted by desert soils, red desertic soils, sierozems, the red and yellow soils of the foothills,
the saline soils of the depressions, and the lithosols (shallow weathered soils) and regosols (soft loose
soils) found in the hills. The subtropical desert climate here results from persistent high pressure and
subsidence. The prevailing southwest monsoon winds from Indian Ocean that bring rain to much of the
Indian subcontinent in summer tend to bypass the Thar to the east. The soils are generally infertile and
overblown with sand due to severe wind erosion (Augustyn et al., 2019). The amount of annual rainfall
in the desert is low, ranging from about 100 mm or less in the west to about 500 mm in the east. Almost
90 % of the annual rainfall occurs in the season southwest monsoon, from July to September. While the
prevailing wind is dry northeast monsoon during other seasons. Dust storms and dust-raising winds are
common from May to July (Chauhan, 2003). Thus, the DUSMASS used in this study is averaged from
June to July and May is neglected to weaken the disturbance of seasonal climatological differences.
Analysis is carried out over the dust source in the northwestern South Asia (65°–82° E, 24°–32° N in Fig.
1). All variables involving spatial average are taken from this region unless stated otherwise.

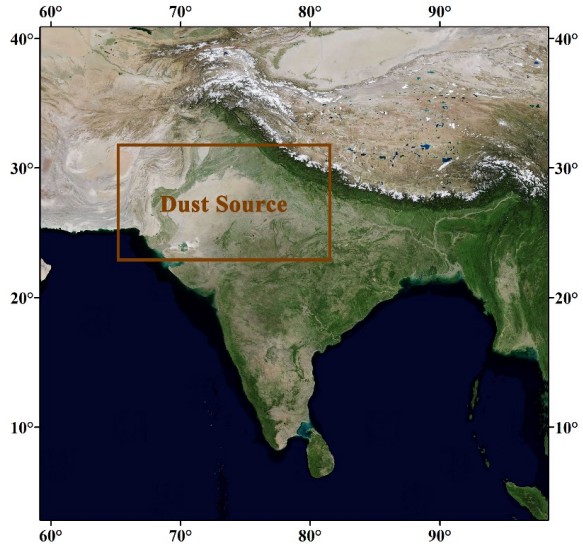




**Figure 1: Geographical map of South Asia (© Google Maps 2021). The dust source over the northwestern South Asia is marked with brown rectangle.**

## 2.2 Datasets

### 2.2.1 Dust concentration

Surface dust concentrations from 1982 to 2014 were provided by Modern-Era Retrospective Analysis for Research and Applications, version 2 (MERRA-2). MERRA-2 is produced via the Goddard Earth Observing System-Data Assimilation System (GEOS-DAS, version 5.12.4) based on GEOS-5 climate model and the Gridpoint Statistical Interpolation (GSI) analysis scheme (Gelaro et al., 2017). Extensive satellite data are integrated into MERRA-2 to estimate dust concentration (Rienecker et al., 2011; Veselovskii et al., 2018). The dust products were comprehensively validated using the results of ground-based observation, satellite measurements, and numerical simulation (Rienecker et al., 2011). They have been widely applied to researches on global environment and climate change (He et al., 2019; Randles et al., 2017). The variable "Dust Surface Mass Concentration" (DUSMASS) with a spatial resolution of 0.625°×0.5° (longitude×latitude) used in this study is from the dataset of "tavgM_2d_aer_Nx".

### 2.2.2 Land and sea surface temperature

To explore the possible influence of SST variability on the South Asian dust activity, we used three SST datasets from 1981 to 2014: (1) The National Oceanic and Atmospheric Administration (NOAA) Extended Reconstructed SST (ERSST) version 5 (Huang et al., 2017) that is available at 2°×2° spatial resolution is used for analysis, (2) Centennial in situ Observation-Based Estimates (COBE) version 2 SST data at 1°×1° spatial resolution (Hirahara et al., 2014) and (3) Hadley Centre Global Sea Ice and Sea Surface Temperature (HadISST1.1) dataset produced by the Met Office, starting from 1870 up to the present with a horizontal resolution of 1°×1° (Rayner et al., 2003) are used for comparison. While the land-sea thermal contrast was calculated from the Hadley Centre Climate Research Unit Temperature version 5.0.1.0 (HadCRUT5) data from 1981 to 2014, which are a blend of the Climatic Research Unit land-surface air temperature dataset (CRUTEM5) and the Hadley Centre sea-surface temperature (HadSST4) dataset (Osborn et al., 2021). The longitude and latitude of SST index involved in this study were shown in Table 1.





**Table 1: Longitude and latitude of SST index used in this study.**

| Acronyms | Full Name | Longitude and Latitude | Involved Ocean |
|---|---|---|---|
| Niño-3 | —— | 150°–90° W, 5° S–5° N | Pacific |
| Niño-3.4 | —— | 170°–120° W, 5° S–5° N | Pacific |
| Niño-4 | —— | 160° E–150° W, 5° S–5° N | Pacific |
| ASGI | Atlantic SSTA gradient index | North: 60°–30° W, 0–20° N<br>South: 20° W–10° E, 0–20° S | Atlantic Ocean |
| TWISSTA | Tropical western Indian ocean SSTA | 50°–70° E, 10° S–15° N | Indian Ocean |
| PDO | Pacific decadal oscillation | 117.5° E–77° W, 20°–66.5° N | Pacific |

**2.2.3 Large-scale climate indices**
Three monthly Niño indices Niño-3, Niño-3.4, and Niño-4 from 1981 to 2014, which monitor the
SST anomalies averaged across the eastern equatorial Pacific, Pacific from dateline to the South
American coast, and central equatorial Pacific, respectively, were used to analyze their links with
DUSMASS over the northwestern South Asia. Kinter et al. (2002) pointed out that Nov.–Jan. is the peak
season for El Niño/La Niña, thus the average Niño index from (–1) Nov. to (0) Jan. was used. Only one
Niño index that shows the highest correlation coefficient was retained in this study, i.e., Niño-3. The
large-scale climate indices, such as PDO, was also used to explore the potential factors that contributed
to the interdecadal shift of ENSO–DUSMASS relationship. All those indices were from the Climate
Predict Center of National Oceanic and Atmospheric Administration (NOAA/CPC).
**2.3 Method**
In this study, we compared the impact of ENSO on DUSMASS over the northwestern South Asia
under two different warming epochs, and investigate the potential global change drivers to the shift of
ENSO–DUSMASS relationship. The global warming is separated into the accelerated warming period
from 1982 to 1996 and the warming hiatus period from 2000 to 2014. The year 2014 was added to the
warming hiatus period to keep the length of those two periods consistent. This classification is not
controversial since the ENSO year stated in this study spanned from antecedent November to current



January.

### 2.3.1 Contribution of factors to relationship

The contribution of Z (Indian Ocean SSTA variance, Atlantic SSTA gradient index, or Eurasian
continent and Indian Ocean thermal contrast) modifying ENSO–DUSMASS relationship was defined as:
sliding regression of Z onto Niño-3 index multiplies by sliding regression of DUSMASS onto Z with
Niño-3 removed (Yang and Huang, 2021).

### 2.3.2 Signal removal method

The residual time series based on the linear regression method were used to remove the ENSO signal
in the oceanic SSTA pattern, (Yang and Huang, 2021), as shown in Eq. (1):

$$\xi_{remove} = \xi - Z \times \frac{cov(\xi, Z)}{var(Z)} \qquad (1)$$

Where $\xi$ and $Z$ is the time series of oceanic SSTA (such as Atlantic and Indian Ocean) and ENSO,
respectively, $cov$ indicated the covariance between two variables, and $var$ indicates the variance of
ENSO.

### 2.3.3 Coupled spatial pattern analysis

The maximum covariance analysis (MCA) is a useful tool for isolating the most coherent pairs of
spatial patterns and their associated time series by performing an eigenanalysis on the temporal
covariance matrix between two geophysical fields (Storch and Zwiers, 1999). The MCA method was
used to analyze the coupled patterns between DUSMASS and oceanic SSTA.

### 2.3.4 Definition of different types of ENSO

Following Yeh et al. (2009), an El Niño event is defined as CP El Niño if Niño-4 > Niño-3 and
Niño-3.4 > 0.5°C, and as EP El Niño if Niño-4 < Niño-3 and Niño-3.4 > 0.5°C. Similarly, a La Niña
event is referred to as CP La Niña if Niño-4 < Niño-3 and Niño-3.4 < –0.5°C, and as EP La Niña if Niño-
4 > Niño-3 and Niño-3.4 < –0.5°C. According to this method, the CP El Niño years during 1982–2014
include 1988, 1991, 1995, 2003, 2005, 2007, and 2010; CP La Niña years include 1984, 1989, 1999,
2001, 2011, and 2012; EP El Niño years include 1983, 1987, 1992, and 1998; EP La Niña years include
1985, 1996, 2000, and 2008.





208  Following Yang and Huang (2021), the EM and CT ENSO were defined based on the Niño-3 index.

209 For CT ENSO, the Niño-3 index follows the rule of $[[(-1)\text{Oct.}-(0)\text{Jan.}]_{mean}>0.5\ (<-0.5)\text{STD}\ \&$

210 $[(0)\text{Mar.}-(0)\text{May}]>0.5\ (<-0.5)\text{STD}\ \&\ [(0)\text{Jun.}-(0)\text{Sep.}]>0\ (<0)]$ or $[[(-1)\text{Oct.}-(0)\text{May}]>0.75$

211 $(<-0.75)\text{STD}\ \&\ [(0)\text{Jun.}-(0)\text{Sep.}]_{mean}>0.5\ (<-0.5)\text{STD}]$. To acquire more available samples in the

212 study period, all the ENSO years that were not defined as CT ENSO were identified as EM ENSO year

213 in this study, which was different from Yang and Huang (2021). Based on this definition, the CT El Niño

214 years during 1982–2014 include 1982, 1983 and 1987; CT La Niña years include 1984, 1985, 1989, 1996,

215 1999, 2000, and 2011; EM El Niño years include 1995, 1998, 2003, 2005, 2007, and 2010; EM La Niña

216 years include 2008 and 2012.

217  The effect of ENSO type on the correlation between ENSO and DUSMASS was partly analyzed

218 through the duration and intensity of Indian Ocean SSTA following the ENSO events. We calculated the

219 correlation between the variance of monthly Indian Ocean SSTA from (–1) Sep. to (0) May and

220 DUSMASS. It is found that the maximum correlation between the Indian Ocean SST and Niño-3 SST

221 occurs in the central Indian Ocean rather than in the Arabian Sea SST (Clark et al., 2000). Therefore, the

222 variance of monthly Indian Ocean SSTA was calculated over the central Indian Ocean (tropical western

223 Indian ocean, TWISSTA) (50–70° E, 10° S–15° N). The EP ENSO tends to onset in spring (Mar.–May)

224 and reaches the mature phase in winter, then decays near Apr. of the second year. Whereas the onset of

225 CP ENSO appears in summer and reaches the peak around Dec. –Jan., then decays in early spring of the

226 second year (Kao and Yu, 2009). Consequently, monthly Indian Ocean SSTA was taken from (–1) Sep.

227 to (0) May.

228  In this study, "(0) month" represents the year concurrent with the year when DUSMASS is acquired

229 and "(–1) month" represents the preceding year.

230 **3 Results**

231 **3.1 Observed interdecadal change of the impact of Niño-3 index on DUSMASS**

232  In the present study, we found that the ENSO–DUSMASS relationship experienced an interdecadal

233 transition at around 1999/2000. Based on the 15-year sliding correlation from 1982 to 2014 (Fig. 3 (b)),

234 the ENSO–DUSMASS relationship was weak before the end of 1990s and became stronger after that.

235 Specifically, the winter Niño-3 index ((–1) Nov.–(0) Jan.) presented a weakly negative relation with





DUSMASS during the accelerated warming period (1982–1996), while exhibited a significant negative
correlation (P < 0.01) during the warming hiatus stage (2000–2014). The varied correlation was not
influenced by the lengthen of sliding windows or the type of Niño index, and was confirmed by multiple
datasets of SST (not shown).
**3.2 Factors influencing the interdecadal change of the impact of Niño-3 index on DUSMASS**
**3.2.1 Tropical Atlantic SSTA pattern**
With the global climate change observed in early 2000s, the ENSO-related tropical Atlantic SSTA
experienced an obvious transition, i.e., from an Atlantic Niña pattern during 1982–1996 to an Atlantic
Niño pattern during 2000–2014 (Fig. 2), which is also reported by Yang and Huang (2021). The tropical
Atlantic SSTA pattern was a crucial factor for the restoration of ENSO–ISMR relationship since the late
1990s (Yang and Huang, 2021), thus, it could also disturb the impact of ENSO on dust activities over the
northwestern South Asia. In order to validate the connection between the Atlantic SSTA and the
DUSMASS–Niño-3 relationship, an Atlantic SSTA gradient index (ASGI) was used to describe the SSTA
pattern shift in the tropical Atlantic, which represented the difference of averaged SSTA between tropical
North Atlantic and tropical South Atlantic (marked by two rectangles in Fig. 2). The Atlantic Niña pattern
develops and is most sensitive to ENSO in spring (Tokinaga et al., 2019), thus the SST averaged from
Mar. to May was used in this section.

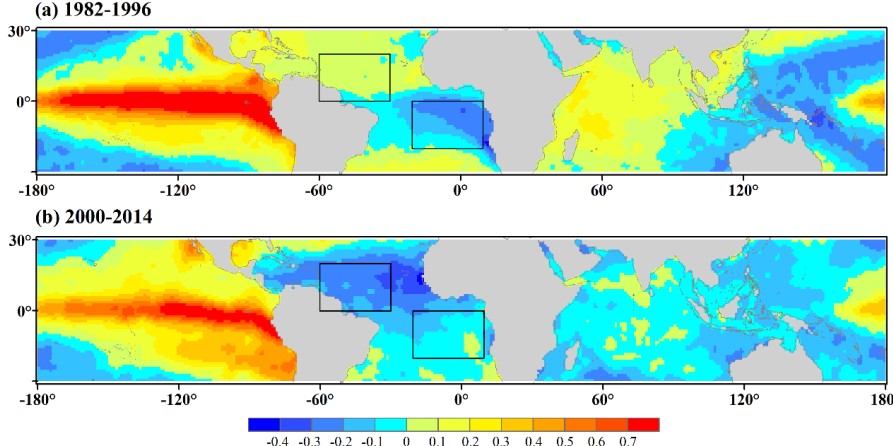


**Figure 2: Regression of tropical SSTA onto Niño-3 index. Black rectangles denote the regions to define ASGI.**
**The range of the upper one is 60–30° W, 0–20° N and that of the lower one is 20° W–10° E, 20–0° S. (Similar**
**with Fig. 2 (c)–(d) of Yang and Huang (2021) but with different time spans)**

Based on the 15-year sliding correlation during 1982–2014 (Fig. 3), the relationship between Niño-

3 and ASGI witnessed a reversal at the early 2000s, simultaneously, the correlation between Niño-3 and

DUSMASS exhibited the similar change. However, the significance of correlation between Niño-3 and

DUSMASS in the two warming phases (1982–1996 and 2000–2014) was opposite to that between Niño-

3 and ASGI, i.e., the correlation between Niño-3 and ASGI passed the 99 % confidence level during

1982–1996 and it did not pass the 95 % confidence level during 2000–2014, while the correlation

between Niño-3 and DUSMASS showed a contrary trend with a higher correlation coefficient appeared

in 2000–2014. Additionally, Figure 4 proved that during 1982–1996, ASGI weakened the DUSMASS–

Niño-3 relationship while the contribution of ASGI was close to 0.0 during 2000–2014. All these clarified

that the weakening of the influence of ASGI on Niño-3 index promoted the effect of Niño-3 index on

DUSMASS.

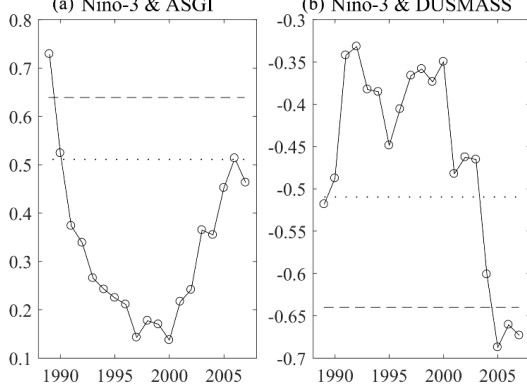


**Figure 3: The 15-year sliding correlation between (a) Niño-3 and Atlantic SSTA gradient index (ASGI), (b)**
**Niño-3 index and DUSMASS during 1982–2014. The x-axis denotes the middle year of the period under**
**analysis. The horizontal dashed line and dotted line indicate the 99% and 95 % confidence levels respectively.**

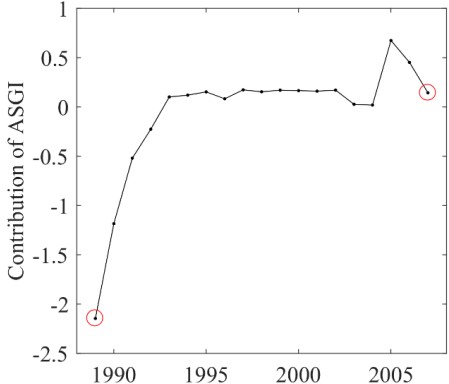




**Figure 4: Sliding contribution of Atlantic SSTA gradient index (ASGI) to ENSO–DUSMASS relationship. The**
**two circles represented the 15-year window spanning from 1982 to 1996 and 2000 to 2014, respectively. The**
**x-axis denotes the middle year of the period under analysis.**

To illustrate the spatial coupling pattern of Atlantic SSTA and DUSMASS, the MCA method was

utilized. Figures. 5 (a)–(b) showed the correlation between DUSMASS and tropical Atlantic SSTA of the
first MCA mode with ENSO-related signals removed. It is clear that the DUSMASS presented
remarkable decrease over northern and northwestern India, meanwhile, the tropical Atlantic SSTA
displayed an apparent dipole pattern with negative in the south and positive in the north.

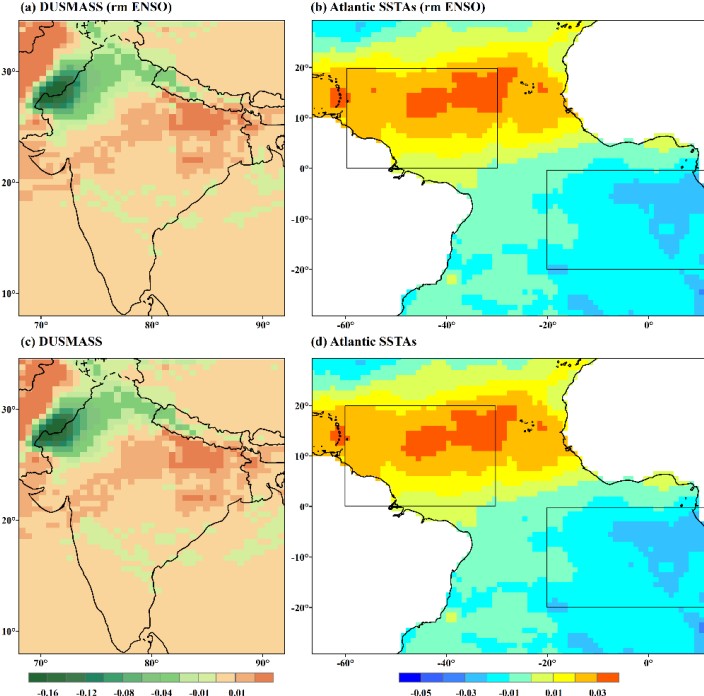


**Figure 5: Spatial correlation between tropical Atlantic SSTA and DUSMASS of the first mode of the MCA**
**analysis in 1982–2014. The first MCA mode of (a) the DUSMASS, and (b) the tropical Atlantic SSTA with**
**ENSO-related signals removed. (c)–(d) As in (a)–(b), but for the original series including the ENSO signal.**

The MCA results with ENSO-related signals removed were similar with that including the ENSO-

related signals (Figs. 5 (c)–(d)), demonstrating that ENSO exhibited no significant impact on the spatial
coupling between DUSMASS and Atlantic SSTA. Besides, the regression of DUSMASS onto ASGI was
close in 1982–1996 and 2000–2014 (not shown), which were both insignificant, suggesting no apparent
interdecadal transition in the relationship between DUSMASS and Atlantic SSTA in recent decades. This
manifested that the shift of the relationship between DUSMASS and ENSO was not due to the



interdecadal transition in the relationship between DUSMASS and Atlantic SSTA.

The MCA result indicated that the colder tropical South Atlantic could suppress dust storm over the

northwestern South Asia but the warmer tropical South Atlantic could enhance the dust concentration.
As reported previously (e.g., Kucharski et al., 2009, 2008, 2007; Kucharski and Joshi, 2017; Sabeerali et
al., 2019; Yadav, 2016, 2009), the warm tropical North Atlantic and clod tropical South Atlantic were
conducive for ISMR, which was bridged by the large-scale monsoon circulation (Rong et al., 2010),
Rossby wave train (Yadav, 2009), Asian jet (Yadav et al., 2018), Kelvin wave response (Sabeerali et al.,
2019), and an abnormal westerly. Considering the relationship between ISMR and tropical Atlantic SSTA,
the Atlantic Niña pattern that was linked to ENSO in 1982–1996 (Fig. 2 (a)) tended to suppress dryness
and weaken the positive relationship between DUSMASS and ENSO, while the Atlantic Niño pattern in
2000–2014 would decrease the north-south Atlantic SSTA gradient and offset the effect of Atlantic Niña
pattern, which weakened the dryness's response to Atlantic and enhanced the ENSO–DUSMASS
relationship. Accordingly, the correlation between ENSO and DUSMASS strengthened together with the
change of Atlantic SSTA pattern.

It was reported that the interdecadal shift of tropical Atlantic SSTA pattern was a response to the

multi-year ENSO events (Tokinaga et al., 2019). The multi-year ENSO event, which was also called as
continuing ENSO (CT ENSO), was a situation where the summer ENSO SSTA continued from the
preceding year. Another type of ENSO, which was called as emerging ENSO (EM ENSO), was
characterized as late Atlantic SSTA response that started from June. The CT ENSO primarily dominated
during 1982–1996, while 2000–2014 was dominated by EM ENSO (Yang and Huang, 2021). The impact
of the two types of ENSO on the shift of the DUSMASS–Niño-3 relationship were examined. Table 2
showed that ASGI was significantly correlated with Niño-3 in the CT ENSO years, which was not
observed in the EM ENSO years. Simultaneously, DUSMASS was significantly related to Niño-3 only
in the EM ENSO years. The composite correlation changes for CT and EM ENSOs resembled that for
ENSOs in 1982–1996 and 2000–2014, indicating that the shift of Atlantic SSTA pattern plays an
important role in modulating the DUSMASS–Niño-3 relationship.
**Table 2: Correlation between ASGI and Niño-3 as well as DUSMASS in two different phases (* and \*\*\***
**indicate the correlations that are significant on a 0.1 and 0.01 level, respectively).**

| R | CT ENSO | EM ENSO | 1982–1996 | 2000–2014 |
| --- | --- | --- | --- | --- |





| | | | | |
|---|---|---|---|---|
| ASGI & Niño-3 | 0.78 (***) | 0.19 | 0.73 (***) | 0.46 (*) |
| DUSMASS & Niño-3 | -0.60 (*) | -0.75 (***) | -0.51 (*) | -0.67 (***) |


Apart from the south-north dipole pattern of the tropical Atlantic SSTA, the switch in the intensity
and location of North Atlantic SST tripole pattern was also responsible for the change of Niño-3–
DUSMASS relationship (Banerjee et al., 2021). Compared with the global warming hiatus period, the
North Atlantic SST exhibited higher correlation with dust activities over South Asia in the context of
global warming, which was teleconnected through the precipitation and westerly anomalies over the
Indo-Gangetic plain (Banerjee et al., 2021). Therefore, the impact of Niño-3 on DUSMASS was
weakened during 1982–1996 when an accelerated global warming was witnessed, while the impact was
stronger during 2000–2014 when a warming hiatus prevailed.
**3.2.2 Variation of Indian ocean SST**
Previous studies have recognized the covariability between the western Pacific and Indian Ocean
(Kug et al., 2005; Wang et al., 2003; Watanabe and Jin, 2002). ENSO can affect the Indian Ocean SST
in the form of Walker circulation and the Indian Ocean variability can also modulate the ENSO variability
(Kug et al., 2005; Wu and Kirtman, 2004; Yu et al., 2002). It is known that ENSO mainly influences the
monsoon rainfall of South Asia through changing the SST of Indian ocean (Krishnamurthy and Kirtman,
2003; Srivastava et al., 2019). During CT ENSO years, the ENSO event in summer primarily starts from
the preceding winter, while in EM ENSO years, the ENSO event mainly emerges in late spring (Yang
and Huang, 2021). Correspondingly, the associated Indian Ocean SST oscillation also varies in these two
different ENSO years. In order to explore whether the different types of La Niña impacted the
DUSMASS over the northwestern South Asia through adjusting the duration of the temperature anomaly,
we compared the SST and the variance of the monthly SSTA from (–1) Sep. to (0) May over the tropical
western Indian ocean (central Indian Ocean, 10° S –15° N, 50–70° E).
Figure 6 showed that the variances in the EM La Niña years were generally larger than that in the
CT La Niña years, while the variances in the EM El Niño years were generally smaller than that in the
CT El Niño years. The monthly tropical western Indian ocean SSTA in CT and EM ENSO years were
seen in Fig. S1, which showed that the monthly SSTA from preceding Sep. to following May appeared
as persistent positive and negative anomalies in EM El Niño and CT La Niña years, respectively, while
those in EM La Niña and CT El Niño years experienced monthly and/or seasonal oscillatory.
Concurrently, the difference of DUSMASS in El Niño and La Niña years was obvious in the EM ENSO
period with higher values appeared in La Niña years (Fig. 7). However, in the CT ENSO period, no
significant difference was observed between El Niño and La Niña years. Therefore, it is hypothesized
that the EM ENSO conditions, which was associated with higher Indian Ocean SST variance, were more
favorable to trigger the variation of DUSMASS. Yang and Huang (2021) reported that 1982–1996 was
primarily dominated by CT ENSOs while EM ENSOs primarily controlled during 2000–2014. Combined
with abovementioned hypothesis, the correlation between DUSMASS and Niño-3 should be higher in
the later period 2000–2014, which was consistent with the interdecadal change of this relationship.

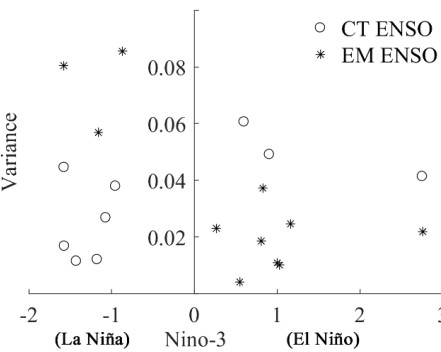


**Figure 6: Scatter diagram between Niño-3 index and the variance of the monthly SSTA from (–1) Sep. to (0)**
**May over the tropical western Indian ocean (50–70° E, 10° S–15° N) separately for continuing (CT) and**
**emerging (EM) ENSO.**

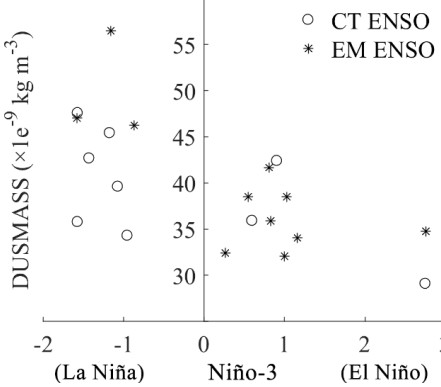


**Figure 7: Scatter diagram between DUSMASS and Niño-3 index separately for continuing (CT) and emerging**
**(EM) ENSO.**
Unlike the weakening effect of tropical Atlantic SSTA pattern on the DUSMASS– Niño-3
relationship, the Indian Ocean SSTA variance, as a response to the type of ENSO, tended to strengthen
this relationship since the impact of Indian Ocean SSTA variance was synchronous with that of Niño-3.
The correlation coefficient between the variance of Indian Ocean SSTA and DUSMASS turned from
0.15 (P>0.1) during 1982–1996 to 0.57 (P<0.05) during 2000–2014, which was different from the
insignificant regression of DUSMASS onto ASGI in those two periods. To quantify the role of Indian
Ocean SSTA variance in modifying the DUSMASS–Niño-3 relationship, we calculated the contribution
of SSTA variance to the relationship as shown in Fig. 8. The contribution of Indian Ocean SSTA variance
was close to 0.0 in 1982–1996 and approached to 1.0 in 2000–2014, which further confirmed the
strengthening effect of Indian Ocean SSTA in the later period.

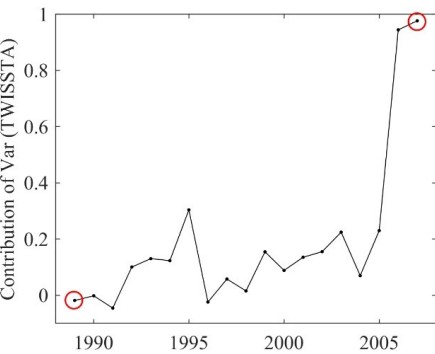


**Figure 8: Sliding contribution of tropical western Indian Ocean SSTA (TWISSTA) variance to ENSO–**
**DUSMASS relationship. The two circles represented the 15-year window spanning from 1982 to 1996 and**
**2000 to 2014, respectively. The Niño-3 index was multiplied by –1 considering that the higher variances in the**
**later period corresponded to the negative phase of ENSO.**

Apart from the CT ENSO and EM ENSO, the response of Indian Ocean SSTA was also different

between CP ENSO and EP ENSO (Hu et al., 2018). It was reported that the onset of EP ENSO appears
in spring and decays near Apr. of the second year, while the CP ENSO tends to onset in summer, reaches
their peak intensity around December–January, and decays in early spring of the second year. The
duration of the CP ENSO is about eight months shorter than that of the EP ENSO (Kao and Yu, 2009).
Especially for the El Niño event, the maximum negative SSTA appears in the winter of year (+1) in the
CP El Niño while a stronger cold anomaly appears in the winter of year (+2) in the EP El Niño, indicating
a slow and strong cooling after the EP El Niño but a quicker and weaker cooling after the CP El Niño
(Hu et al., 2012). In order to explore whether the different types of ENSO impacted the DUSMASS over
northwestern South Asia through adjusting the Indian Ocean SST, we compared the SST and DUSMASS



in CP and EP ENSO conditions.

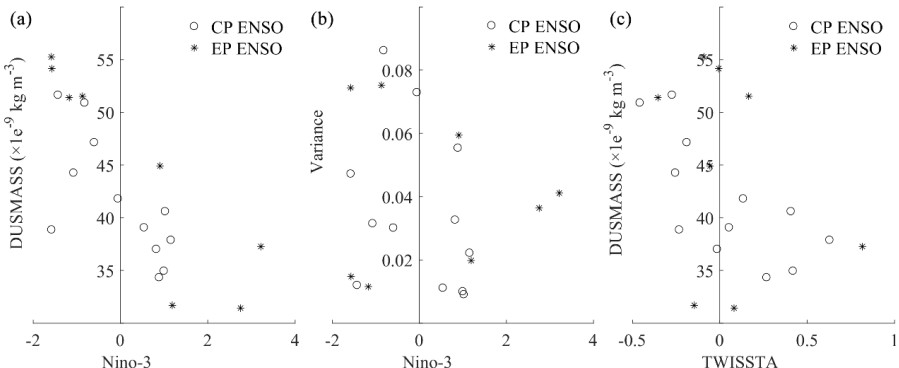

**Figure 9: Scatter diagram between (a) Niño-3 and DUSMASS, (b) Niño-3 and TWISSTA variance, and (c) TWISSTA and DUSMASS separately for CP and EP ENSO.**

Figure 9 (a) showed that the DUSMASS was generally larger under La Niña conditions than that under El Niño conditions. Specifically, the mean DUSMASS in EP La Niña years was higher than that in CP La Niña years, however, this difference was not significant. In addition, no remarkable difference in Indian Ocean SSTA variance was observed under the two types of ENSO, as shown in Fig. 9 (b). The correlation analysis (not shown) revealed that the significant interdecadal variability of Niño-3– TWISSTA and DUSMASS–TWISSTA relationship was spotted in winter (averaged from (–1) Dec. to (0) Feb). Nevertheless, the TWISSTA under two types of ENSO and the DUSMASS in varied TWISSTA exhibited no significant differences (Fig. 9 (c)), indicating that the variation of TWISSTA and the response of DUSMASS to TWISSTA was not modulated by the types of EP and CP ENSO. The results demonstrated that the transition of EP and CP ENSO exhibited no significant contribution to the shift of the DUSMASS–Niño-3 relationship.

### 3.2.3 Eurasian continent and Indian Ocean thermal contrast

Kumar et al. (1999) pointed out that in El Niño events before 1980, colder Eurasian temperature anomalies coupled with positive Pacific SST anomalies coincides with negative ISMR anomalies. The inverse connection between ENSO and Indian summer monsoon is maintained. While in the El Niño events during 1981–1997, the increased premonsoon surface temperatures over Eurasia far exceeded the warming in the Indian Ocean. The stronger lad-sea thermal contrast is conducive for a stronger monsoon, which overrides the influence of El Niño. The Eurasian temperature anomalies could also disturb the impact of ENSO on DUSMASS over the northwestern South Asia since monsoon and rainfall are crucial



factors to trigger dust storms. To isolate the role of Eurasian continent temperature anomalies in
modulating the effect of ENSO on DUSMASS over the northwestern South Asia in the recent climate
change (i.e., from an accelerated global warming during 1970s–1990s to a warming hiatus during 1998–
2014), we compared the Eurasian continent (60°–100° E, 30°–45° N) and Indian Ocean (60°–100° E, 10°
S–10° N) thermal contrast in those two periods and calculated its contribution to the change of ENSO–
DUSMASS relationship.

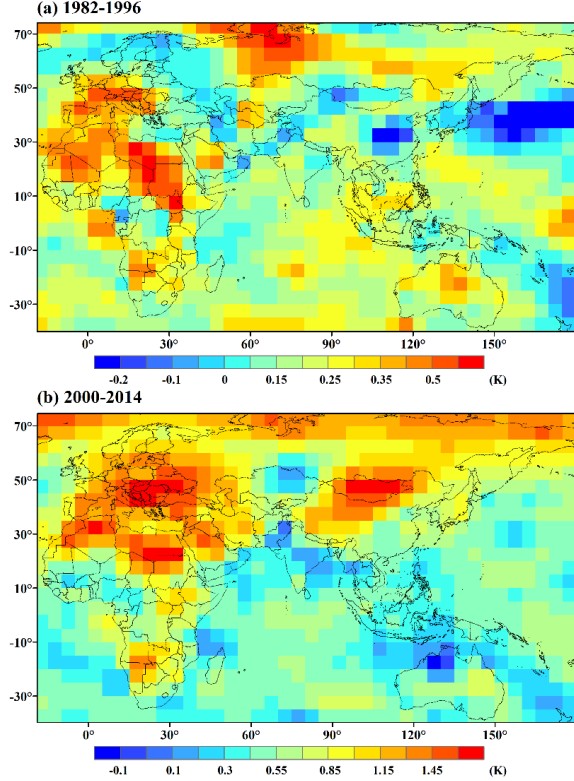


**Figure 10: Spatial distribution of summer surface temperature anomaly separately for 1982–1996 (a) and**
**2000–2014 (b).**

Figure 10 displayed the spatial distribution of summer surface temperature. Compared to the spatial

distribution of surface temperature during 1982–1996, the warming over Eurasian continent was
significantly stronger than that over tropical Indian Ocean during 2000–2014, which indicated higher
land-sea thermal contrast in the later period. The strong land-sea thermal contrast can significantly impact
the drought conditions over the arid regions in the northwestern South Asia through adjusting the south
Asian monsoon, moisture transmission, and precipitation (Kinter et al., 2002; Kumar et al., 1999).

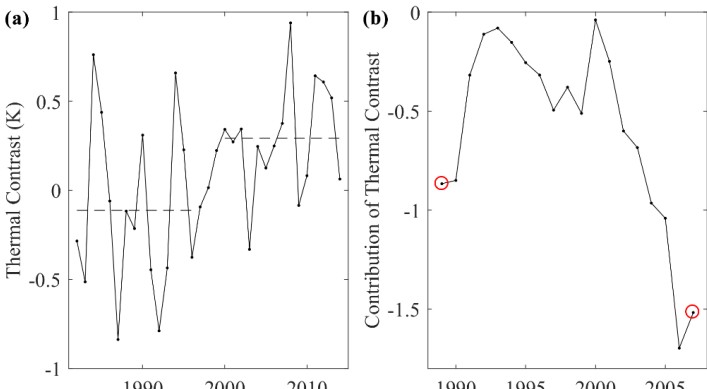


**Figure 11: Time series of summer Eurasian continent and Indian Ocean thermal contrast. The dotted lines**
**denote the average values from 1982 to 1996 and from 2000 to 2014, respectively; (b) Sliding contribution of**
**Eurasian continent and Indian Ocean thermal contrast to ENSO–DUSMASS relationship. The circles**
**represented the 15-year window spanning from 1982 to 1996 and from 2000 to 2014, respectively.**
As shown in Fig. 11 (a), the Eurasian continent and Indian Ocean thermal contrast during 2000–
2014 was higher than that in 1982–1996. As a result, the negative contribution of the thermal contrast
grew stronger in the later period compared with that in the first period. The enhancement of thermal
contrast's effect mitigated the impact of Niño-3 index on DUSMASS, and this should have led to higher
correlation coefficient in 1982–1996 rather than in 2000–2014, which conflicted with the observed
interdecadal transition of DUSMASS–Niño-3 relationship. Thus, the change of Eurasian continent and
Indian Ocean thermal contrast cannot explain the difference of DUSMASS–Niño-3 relationship in the
varied warming phases.
**3.2.4 Phase shift of Pacific Decadal Oscillation**
It is suggested that the PDO can influence the interannual variability of ISMR by enhancing the
ENSO–ISMR relationship when ENSO and the PDO are in-phase, while weakening the relationship
when they are out of phase (Dong et al., 2018; Krishnamurthy and Krishnamurthy, 2014). However, it is
unclear whether the PDO is responsible for the shift of the ENSO–DUSMASS relationship. Table 3 listed
the years with different phases of ENSO and PDO as well as years when ENSO and PDO are in (out of)
phase separately. The correlation coefficient between Niño-3 and DUSMASS and significance level were
also given. It demonstrated that the PDO significantly strengthened the correlation between Niño-3 and
DUSMASS as the coefficient turned from –0.34 (P>0.1) when PDO and Niño-3 were out of phase to –
0.71 (P<0.01) when they were in-phase.



The 200 hPa velocity potential trend in the positive PDO years exhibited a decrease (divergence)
over the western tropical Pacific and an increase (convergence) over the tropical Indian Ocean and Indian
subcontinent (across 40°–100° E), as shown in Fig. 8 (a) of Huang et al. (2020). The upper-level
convergence over India and the adjacent seas corresponded to the anomalous descending motion, which
suppressed ISMR and consequently stimulated the dust storms over South Asia. Meanwhile, two
anomalous anticyclones developed to the northwest and southwest of India due to the depressed
convection (Huang et al., 2020). The westerlies on the northern flank of the northwest anticyclonic
anomalies advected relatively drier air from the Eurasian continent (Parker et al., 2016) to the west of
north-central India. Those situations were in favor of dust generation, in addition, the eastward transport
of dust from Southwest Asia by the westerlies contributed about half of the dust concentration over the
Indo-Gangetic plain (Banerjee et al., 2019). While in the negative PDO years, anomalous ascent appeared
over India, which enhanced ISM convection and rainfall, as shown in Fig. 8 (b) of Huang et al. (2020).
Similarly, two anomalous cyclones established to the west of India. The easterlies over the northern
Indian subcontinent transported wet air from the Bay of Bengal into the east of north-central India, which
increased rainfall and reduced dust emissions over the northern India (Huang et al., 2020). The
descending and ascending flows over the Indian subcontinent in the negative and positive phase of PDO
coincided well with that in the La Niña and El Niño periods, respectively. Hence, PDO can significantly
strengthen the effect of ENSO when they were in-phase.
**Table 3: List of individual and combined wintertime ENSO–PDO years during 1982–2014.**

| Events | Phase | |
|---|---|---|
| | Positive | Negative |
| ENSO | 1983, 1987, 1988, 1991, 1993–1995, 1998, 2003, 2005, 2007, 2010, 2015, 2016, 2019 | 1982, 1984–1986, 1989, 1996, 1997, 1999–2001, 2006, 2008, 2009, 2011, 2012, 2014 |
| PDO | 1981–1988, 1996–1998, 2001, 2003–2006, 2010, 2014–2019 | 1989, 1991, 1995, 1999, 2000, 2002, 2008, 2009, 2011, 2012 |
| ENSO×PDO | 1983, 1987–1989, 1998–2000, 2003, 2005, 2008–2012 | 1982, 1984–1986, 1991, 1995–1997, 2001, 2006, 2014 |





| R (Niño-3 & DUSMASS) | –0.71 (P<0.01) | –0.34 (P>0.1) |
|---|---|---|

Table 3 revealed that most of years (8 out of 14) when ENSO and PDO were in-phase appeared in
the second period, i.e., 2000, 2003, 2005, and 2008–2012. Meanwhile, most of years (8 out of 11) when
ENSO and PDO were out of phase appeared in the first period, i.e., 1982, 1984–1986, 1991, 1995, 1996,
and 1997. Simultaneously, the winter Niño-3 exhibited lower correlation with DUSMASS in the first
period when most of ENSO years were accompanied with anti-phase PDO. In addition, the quantitative
contribution of PDO shown in Fig. 12 further confirmed that the PDO strengthened the impact of ENSO
on DUSMASS in 2000–2014 while the contribution was close to 0.0 in 1982–1996. All those
demonstrated that the phase shift of PDO plays an important role in modulating the revolution of
DUSMASS–Niño-3 relationship.

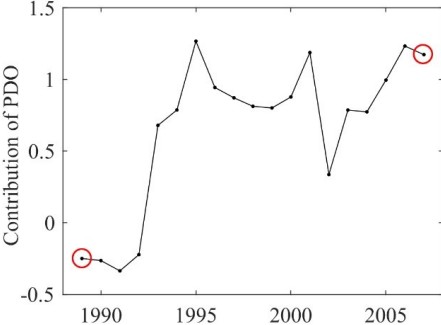


**Figure 12: Sliding contribution of PDO to ENSO–DUSMASS relationship. The two circles represented the 15-**
**year window spanning from 1982 to 1996 and 2000 to 2014, respectively.**
**4 Discussion**
**4.1 Uncertainty in analyzing the contribution of the influence factors**
The schematic diagram of the interdecadal shift of the ENSO impact on DUSMASS was shown in
Fig. S2. It illustrated that all the SSTA patterns in Atlantic Ocean, Pacific Ocean, and Indian Ocean play
important roles in changing the ENSO–DUSMASS relationship. The contributions of those
abovementioned factors to the interdecadal shift of this relationship were analyzed based on the linear
regression model. However, the linear regression model would definitely bring uncertainty to the results
(Guo et al., 2017) and may not be sufficient to verify the causal relationship between the factors and the
ENSO–DUSMASS relationship. The numerical models are thus suggested to more accurately quantify





the contribution of those factors to the shift of ENSO–DUSMASS relationship, which will be included
in our future research. However, it is undeniable that this study provides new sights to the dust storm-
related numerical simulation by taking account of the teleconnections and their influence mechanisms.
In addition, while analyzing the effects of different types of ENSO event in Sect. 3.2.2, we compared
the variance of the difference in SST over the tropical western Indian ocean between two adjacent months,
as shown in Figs. 6, 7, and 9. It showed that only eight EP and EM ENSO years were identified and the
number of CP and CT ENSO years were also insufficient. The statistical results acquired from the
insufficient number of samples could also be explained by the random events (Pallikari, 2004). In order
to verify this conclusion, we calculated the interannual correlation between the variance of SST
difference and DUSMASS from 1982 to 2014. Even so, the significant interannual correlation does not
guarantee the significant link between different types of ENSO. Therefore, longer time series with valid
samples (EP, CP, CT, and EM ENSO years) are needed to further validate the influence of ENSO types
on the ENSO–DUSMASS relationship in the future. Alternatively, using numerical model to simulate
the teleconnection pattern of ENSO over South Asia under different types of ENSO is also favorable.
It is possible that the interdecadal variability of ENSO–DUSMASS relationship results from
anthropogenic land-use management (Kumar et al., 1999), which should be considered in the future
researches.

**4.2 Difference in varied seasons and ENSO types**

In this study, we analyzed the effect of wintertime ((–1) Nov.–(0) Jan.) ENSO on DUSMASS. It is
noteworthy that the correlation between summertime ((–1) Jun.–(–1) Aug) ENSO and DUSMASS also
exhibited remarkable interdecadal variability in the context of global warming, i.e., the correlation
coefficient shifted from –0.24 (P>0.1) during 1982–1996 to –0.81 (P<0.001) during 2000–2014. It is
known that lead-lag interaction and feedback are common among the large-scale atmospheric
teleconnections, and the teleconnection pattern in other oceans usually lags behind the atmospheric
circulation of Pacific Ocean (Trenbeth et al., 2002). Regarding this, the summertime ENSOs were
obtained from the antecedent year, i.e., (–1) Jun.–(–1) Aug. The impact of summer and winter ENSO on
the following summer DUSMASS would be confused when the lag effects were taken into account.
Therefore, only wintertime ENSO impacts were displayed in this study, which could be the superimposed
effects of summertime and wintertime ENSO. The precise numerical simulation is needed to further



isolate the seasonal difference in detail.
The CT ENSO refers to the summer ENSO that primarily starts from the preceding winter, whereas
the EM ENSO refers to the ENSO event that emerges in late spring (Yang and Huang, 2021). Similarly,
the onset of EP ENSOs tends to occur in spring while that of CP ENSOs appear in summer. Besides, both
CT and EP ENSOs dominate in 1979–1997, while the EM and CP ENSOs become more frequent in
2000–2018 (Kao and Yu, 2009; Yang and Huang, 2021). Thus, it is likely that the CT (EM) ENSOs may
coincide with EP (CP) ENSOs. However, the years defined as CT (EM) ENSO years using Yang and
Huang (2021)'s method were not consistent with the EP (CP) ENSO years and they were crossed among
different types of ENSOs, indicating that the CT (EM) ENSOs were independent of the EP (CP) ENSOs.
Yang and Huang (2021) suggested that the impact of ENSO on ISMR is significant when EM ENSOs
dominated, while it is weak when CT ENSOs prevailed. It is further demonstrated that the transition of
EP and CP ENSO exhibits no significant contribution to the shift of relationship between ISMR and
ENSO since the 21st century. Those are also true for the revolution of ENSO–DUSMASS relationship.
**5 Conclusions**
In the study, we investigated the interdecadal change of the ENSO impact on DUSMASS over the
northwestern South Asia from 1982 to 2014 as well as factors that contribute to the shifts of the responses
of DUSMASS to the wintertime ENSO. It was found that the ENSO–DUSMASS relationship shifted
from weak negative relation during 1982–1996 to significant negative correlation during 2000–2014.
The change of Atlantic SSTA pattern weakened the impact of wintertime ENSO on dust activities over
the northwestern South Asia, while that of Indian Ocean SSTA pattern and PDO tended to strengthen
ENSO's effect. Both the Atlantic and Indian Ocean SSTA patterns were modulated by the duration of
ENSO events (i.e., continuing and emerging ENSO). The Eurasian continent and Indian Ocean thermal
contrast cannot explain the shift of ENSO–DUSMASS relationship. The current results are based solely
on the linear regression, and further studies integrating numerical models and longer time series are
needed to validate the results. Nevertheless, our study indeed found multiple large-scale factors that could
impact the interdecadal interaction between ENSO and dust activities over the northwestern South Asia.
Considering the large-scale circulation forecast is relatively easier, the results in this study could provide
new insight to the prediction of dust storm trend in the near future based on the variability of ENSO–





DUSMASS relationship.
**Data availability**
The data can be downloaded for free from the corresponding website which were listed in the text.
**Author contribution**

549         L.S. designed the study, performed the analysis with feedback from J.Z. and F.Y., and wrote the

paper that was reviewed by J.Z., F.Y., D.Z., J.W., X.M., and Y.L.. All the authors discussed the results.
**Competing interests**

The authors declare that they have no conflict of interest.

**Acknowledgments**

The authors would like to thank the Modern-Era Retrospective Analysis for Research and

Applications, version 2 (MERRA-2) for providing the surface dust mass concentration, wind speed and
planetary boundary layer height, the National Oceanic and Atmospheric Administration (NOAA) for
providing the SST, the Hadley Centre Climate Research Unit for the land-surface temperature and
precipitation, the National Aeronautics and Space Administration (NASA) for the NDVI, and the Climate
Predict Center of National Oceanic and Atmospheric Administration (NOAA/CPC) for the large-scale
climate indices.

This work was supported by the Strategic Priority Research Program of the Chinese Academy of

Sciences-A (No. XDA19030402) and the National Natural Science Foundation of China (No. 42071425).

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
