# Peer review of "What caused the interdecadal shift of the ENSO impact"

_Atmospheric Chemistry and Physics, 2022_

## Author Comment (AC1)

To reviewer #1

Dear Professor:

Thank you for your kind comments for our manuscript entitled "**What caused the interdecadal shift of the ENSO impact on dust mass concentration over northwestern South Asia**", submitted to Atmospheric Chemistry and Physics. We appreciate your valuable comments and suggestions to improve it. We are sincerely grateful for giving us the opportunity to improve our work. With regard to your comments and suggestions, we wish to reply as follows:

**Major concerns:**

Comment 1: The most concerning point is that this study only investigates the regression/correlation type of relationship between large-scale modes of climate variability, e.g. SSTAs, and dust. In absence of any atmospheric and land surface drivers (e.g. precipitation, wind, soil moisture, vegetation cover, etc) of dust, it is hard to believe any causal link between remote SSTAs and regional dust concentration. While I notice the authors have cited many previous studies, I don't feel their results are directly applicable to your scientific questions, due to different analyzed datasets, time periods, etc.

**Response:** Thank you so much for pointing out this to help us to perfect the results. The absence of atmospheric and land surface drivers of dust is a great drawback of this study, thus we added the influence mechanism of Atlantic SSTA pattern, Indian ocean SSTA pattern, and PDO on the relationship between ENSO and dust concentration by analyzing the effects of those factors on convection, precipitation, wind, NDVI, and soil moisture.

More specifically, for the influence mechanism of Atlantic SSTA index (ASGI), we compared the effect of North and South Atlantic SSTA with ENSO signal removed on the geopotential height at 850hPa/300hPa and zonal/meridional wind at 300hPa during the two study periods (Figs. 6-7), the results indicated that the response of atmospheric circulation on North and South Atlantic SSTA index modulated the ENSO-dust relationship. Besides, we compared the effect of ASGI and ENSO on local climate elements (soil moisture, precipitation, NDVI, wind field, and velocity potential) to prove that ASGI and ENSO exhibited the opposite effect on dust activities in the first period (Figs. 8-9).

As for the influence mechanism of Indian ocean SSTA index (TWISST), we compared the effect of TWISST and ENSO on local climate elements (soil moisture, precipitation, NDVI, wind field, and velocity potential) to prove the impact of TWISST on the ENSO-dust relationship (Figs. 12-14).

While for the influence mechanism of PDO, originally, we cited others' results to explain the influence mechanism, however, according to your instructive suggestion, we did the analogous analysis based on the datasets and time periods of this study. We compared the convection and its effect on dust activities between negative and positive phase of PDO as well as that between negative and positive phase of ENSO, the analogous effect of in-phase PDO and ENSO proved that PDO can strengthen the

impact of ENSO on dust activities when it was in phase with ENSO (Fig. 16).

Comment 2: In this study, the authors investigated surface dust mass concentration from reanalysis. My understanding is that reanalysis covers longer period than satellite aerosol products. But it is worthy of checking the quality of the reanalysis product over the study area, with ground observations (AERONET) (Holben et al., 2001), direct remote-sensing product (e.g. MISR nonspherical AOD) (Garay et al., 2020), particulate matter concentrations (PM10) (Yu et al., 2021), visibility and weather observations from weather stations (Xi, 2021). I guess another purpose of analyzing surface dust concentration is to focus on locally emitted dust, rather than complicating the result interpretation with transported dust from for example Arabian Peninsula. But I suspect the uncertainty of reanalysis surface dust concentration is higher than columnar dust concentration, because the former requires an accurate representation of dust vertical distribution beyond the total dust amount required by the latter metric.

**Response:** Thank you so much for pointing out this to help us to improve our work. It is true that the uncertainty of reanalysis surface dust concentration is higher than dust column concentration. We compared the time series of monthly surface dust mass concentration (DUSMASS) with that of dust column concentration (DUCMASS), to find that these two datasets showed consistent variation trend. In the revised manuscript, we substituted DUSMASS dataset with DUCMASS dataset. Besides, we added the precision validation of MERRA-2 dust concentration dataset in the discussion section 4.4. In Sect. 4.4, we compared the dust emission (DUEM), DUSMASS, DUCMASS from MERRA-2 with coarse mode aerosol optical depth acquired from AERONET, nonspherical aerosol optical depth retrieved from MISR, and $PM_{2.5}$ in a taylor diagram, as shown in Fig. 18. The $PM_{2.5}$ is used because the dust concentration dataset used in this study is "DUCMASS25". The precision validation results indicated that the interannual variations of DUSMASS and DUCMASS were consistent, while they were different from that of DUEM. The time series of DUSMASS and DUCMASS over the dust source region were significantly correlated with $PM_{2.5}$ and those over each AERONET station were closely correlated with the coarse mode aerosol optical depth over the corresponding station. Thus, it is hypothesized that the DUCMASS dataset is reliable to support this study.

Comment 3: The whole manuscript is not well-organized. The main objective of this paper is to test different hypothesized modulators of the ENSO-dust relation on the decadal scale; those hypothesized modulators include Indian Ocean SST, Atlantic Ocean SST, land-ocean thermal contrast, rapid warming versus slow warming, different ENSO types, etc. I personally feel confused about the role of all these different factors. Maybe a schematic diagram illustrating the key findings will help. I would also recommend reorganize the whole manuscript, including introduction and results sections to clearly present these hypotheses and testing results.

**Response:** Thank you so much for this instructive suggestion. We are so sorry for the confusing structure, we reorganize the whole manuscript according to your suggestions.

For the results section, we mainly presented the effect of Atlantic SSTA index, Indian ocean SSTA index, and PDO on the change of ENSO-dust relationship. Simultaneously, according to another reviewer's comments, we deleted the part related to factors insignificant for modulating the ENSO-dust relationship, i.e., the section discussing the effect of Eurasian continent and Indian Ocean thermal contrast. For the responses to ENSO types, we move them to the discussion section (Sect. 4.1 and Sect. 4.2) to make the results section clearer.

For the introduction section, we deleted some unrelated sentences and rewrote it following below outline: 1. Introduce the adverse impacts of dust events and the significance of dust-related researches. 2. Simply review the impact of ENSO on dust activities and the regulatory factors to ENSO's effects. 3. Introduce the interdecadal change of earth climate and the ENSO teleconnection. 4. Summarize the deficiencies of existing studies and the goal of this study.

Comment 4: I'm not sure the sample size allows to draw any conclusions regarding decadal shift in ENSO types. Moreover, the emerging and continuing ENSO events are particularly confusing. From the definition of these events, for example [(0)Mar.–(0)May]>0.5 (<–0.5)STD, did you mean that in continuing El Niño event, March Niño-3 should be greater than May Niño-3 for 0.5 standard deviation? Then why do you call it continuing, given that the SST anomaly is decaying fast? I did not find significant differences between the currently examined EM and CT ENSO years from their SST evolutions provided by CPC for example (https://origin.cpc.ncep.noaa.gov/products/analysis_monitoring/ensostuff/ONI_v5.php).

**Response:** Thank you so much for proposing this, it is worth of consideration since the sample size is really too small. We discussed this deficiency in the section 4.3.

As for the definition of emerging and continuing ENSO events, we are so sorry to give the confused introduction. The definition of these two types of ENSO and the characteristics of emerging and continuing ENSO referred to the statement of Yang and Huang (2021). We modified this introduction as "Following Yang and Huang (2021), the EM and CT ENSO were defined based on the three-month running mean of the Niño-3 index. Two situations for the CT ENSO were considered, i.e., the slowly decaying events and the developing events since the previous winter. For the slowly decaying situation, a CT ENSO was identified when the average Niño-3 of (–1) Oct.–(0) Jan. was greater than 0.5 (below –0.5) standard deviation (STD), became greater than 0.5 (below –0.5) STD in single month during (0) Mar.–(0) May, and remained positive (negative) during (0) Jun.–(0) Sep.. For the developing events since the previous winter, a CT ENSO was identified when the Niño-3 was greater than 0.75 (below –0.75) STD in any month from (–1) Oct. to (0) May, accompanied by positive (negative) values for eight single months, and the average Niño-3 of (0) Jun.–(0) Sep. was greater than 0.5 (below –0.5) STD. To acquire more available samples in the study period, all the ENSO years that were not defined as CT ENSO were identified as EM ENSO year in this study, which was different from Yang and Huang (2021)". There should be no ambiguity in the revised definition.

Comment 5: The sliding regression analysis is a smart way to analyze the relatively short data record. My understanding is that you can identify changing point from the sliding regression. But it might not be necessary to show all the sliding regression results in the main text. I personally would replace those panels with more in-depth investigation of the mechanisms underlying the teleconnections, e.g. response in precipitation, wind, soil moisture, vegetation to those SSTAs.

**Response:** Thank you for your suggestions, it would be better to replace those sliding regression graphs with mechanism analysis. We removed some sliding correlation graphs and added the spatial distribution of correlation between ENSO and dust concentration in the two periods separately (Fig. 2). We also added the influence mechanisms of SSTA pattern on this relationship by analyzing their effects on convection, precipitation, wind, and soil moisture, as shown in Figs. 6-9, 10-14, and 16. However, we kept two sliding regression maps (Figs. 4 and 10), because we aimed to show the synchronous interdecadal changes of the relationship between ENSO and DUCMASS with that between ENSO and oceanic SSTA index. Compared with other display form, we think that the sliding regression graphs are more appropriate to show this interdecadal change.

Comment 6: I noticed that there has never been a climatological dust concentration map or a regression map of dust onto ENSO.

**Response:** Thank you for pointing out this. It would be more intuitive to show the features of the study area using the climatological dust concentration map, thus we added this map, as shown in Fig. 1. In addition, the regression map of dust onto ENSO was also added to illustrate the change of the relationship between ENSO and dust in the two periods, as shown in Fig. 2.

**Minor concerns:**

Comment 1: On line 37, dust can travel, not dust storm.

**Response:** We are so sorry for making this mistake. We changed the "dust storm" into "dust" (line 36).

Comment 2: On line 55, it is not accurate to say that "global warming came to an end in 2013", almost everyone agrees that global warming is continuing.

**Response:** We are so sorry for making this mistake. It should be "global warming hiatus came to an end". It can be deduced from the preceding sentence, i.e., we stated that an accelerated global warming prevailed before late 1990s and a warming hiatus dominated after that, and next, it should be a new phase "end of global warming hiatus". We are so sorry for not checking this carefully and we have changed it into "global warming hiatus came to an end" (line 71).

Comment 3: On line 79, while in the introduction you mention the role of IOD. I suspect

Indian Ocean Basin mode would have an effect (Yang et al., 2007). Why don't analyze IOD and Indian Ocean Basin mode separately, rather than regional average SSTAs as in the current analysis?

**Response:** Thank you so much for pointing out this to help us to improve our work. It is true that IOD plays an important role in the interannual change of dust activities. Thus we explored its effect on interdecadal change of the ENSO-dust relationship in this study, to find that its effect was much weaker that of tropical western Indian ocean SST (TWISST). We also illustrated the reason of choosing TWISST, as stated in the second paragraph of Sect. 3.2.2. They were "Figure 10 (a) showed that the correlation between Niño-3 and DUCMASS was obviously reduced when the tropical western Indian ocean SSTA (TWISST) was removed from Niño-3. The contribution of TWISST to this relationship also illustrated that during P1, TWISST weakened this relationship while no significant contribution was observed during P2, as shown in Fig. 10 (b). Thus, it is hypothesized that TWISST weakened the impact of ENSO on DUCMASS during P1. However, when the IOD (rather than TWISST) was considered, the correlation between Niño-3 and DUCMASS kept the same when IOD was removed from Niño-3, indicating that IOD exhibited no significant impact on the correlation between Niño-3 and DUCMASS. Clark et al. (2000) showed that the SST in the central Indian Ocean exhibited stronger correlation with the Indian precipitation than that in the Arabian Sea and northwest of Australia. Cherchi and Navarra (2013) also pointed out that when the eastern and western poles of the IOD were considered separately, the western side exhibited the largest correlation. Thus, the TWISST was considered when exploring the effect of Indian ocean SSTA pattern on the DUCMASS–Niño-3 relationship." (Lines 316-328)

Comment 4: Lines 86-88, this sentence does not flow well. The first part talks about ENSO impacts on DUSMASS due to its influence on winter precipitation, the second part talks about interdecadal change in ENSO-Indian Summer Monsoon Relationship.

**Response:** Thank you so much for pointing out this to help us to perfect our writing. We are so sorry for this confused sentence. We rewrote this sentence since we reorganize the structure of the introduction section. We have changed this sentence into "It was reported that the effect of ENSO on Indian summer monsoon rainfall (ISMR), which was an important modulator to DUCMASS, experienced a remarkable interdecadal change and many factors may cause this transition (Yang et al., 2021). Till now, the interdecadal variability in the links of DUCMASS over the northwestern South Asia with ENSO has not been fully investigated, compared with the North African and West Asian counterpart". (Lines 83-87)

Comment 5: Lines 98-99, AOD cannot be provided by meteorological stations.

**Response:** We are so sorry for this wrong expression. What we intend to show here is ground-based observation stations rather than meteorological stations. However, we have deleted this sentence since we have reorganized the introduction section.

Comment 6: Lines 129-133, do you mean that the prevailing wind shifts from May to

June? But the above sentence states that the July to Sep is the summer monsoon season. Then why do you focus on June to July?

**Response:** Thank you for pointing out this, we are so sorry for the confused statement. It is known from previous references that the dust activities over the study area were most active during May-July. We emphasized May-July here because this study focused on the dust activities in the dust season. However, based on the climatological definition, May belongs to spring, and June-July belong to summer. Considering that the atmospheric and climate conditions exhibit great differences in different seasons, this study separated May from June-July to eliminate the influence of seasonal climatological differences. Besides, one more important point is that May belong to pre-monsoon season and June-July belong to monsoon season (Babu et al., 2013). Our results indicated that the significant interdecadal change of the ENSO-DUCMASS relationship occurred only when the DUCMASS during dust season was considered, which was not seen during pre-monsoon season. Thus, this study focused on June to July. The difference of the ENSO impact on DUCMASS between pre-monsoon season and monsoon season were discussed in Sect. 4.3 (Lines 478-485).

Comment 7: Figure 1, I recommend replacing with a mean DUSMASS map.

**Response:** Thank you so much for this instructive suggestion. We changed Fig. 1 into a map of climatological mean DUCMASS (Lines 122-124).

Comment 8: Line 147, this reference is probably too old for MERRA2, which came out in 2017.

**Response:** Thank you so much for pointing out this to help us to perfect our manuscript. Actually, this reference (Rienecker et al., 2011) is for MERRA rather than MERRA2. We are so sorry for making this mistake. We searched articles about the application and validation of MERRA2 dust products and replaced "(Rienecker et al., 2011)" with "(Buchard et al., 2017; Randles et al., 2017)". (Lines 133-134)

Comment 9: In section 2.3.1, Z refers to ocean SSTAs, while in 2.3.2, Z refers to ENSO.

**Response:** We are so sorry for this negligence. It is true that Z represents different variables in section 2.3.1 and section 2.3.2. Thus we changed the "Z" in section 2.3.1 into "X". (Lines 172-174)

Comment 10: Line 234, why don't you show a map of correlation between DUSMASS and Niño-3 for before 1990s and afterwards? And maybe MCA of Pacific SST and DUSMASS.

**Response:** Thank you so much for this instructive suggestion. We added a map showing the correlation between DUCMASS and Niño-3 in the two periods, as shown in Fig. 2. (Lines 215-217)

Comment 11: Table, the sample size is too small.

**Response:** Thank you so much for pointing out this, it is true that the sample size is too small. We discussed the shortage of the sample size in the discussion section 4.3.

We acknowledged that the significant difference, which was acquired based on the insufficient samples, did not guarantee the significant correlation between dust and the large-scale atmospheric circulation pattern. Thus, longer time series with valid samples are needed to further validate the related conclusion. (Lines 464-474)

Comment 12: Lines 422-423, I would not be surprised if ISMR is affected by land-sea thermal contrast over Tropical Indian Ocean and Indian subcontinent, but not Europe and Arabian Peninsula.

**Response:** Thank you so much for pointing out this. It is reported by Kumar et al. (1999) that the change in surface temperature over Eurasia adjusts the land-ocean thermal gradient, which modulates the strong monsoon that influences the correlation between ENSO and ISMR. However, the results of this study indicated that Eurasian continent and Indian Ocean thermal contrast was less likely to influence the interdecadal of ENSO-dust relationship, thus we deleted this subsection according to another reviewer's comments.

Comment 13: Line 540, I think a longer time series is crucially needed for this type of analysis. Maybe you need to first obtain the atmospheric and land surface regulator for dust, then analyze a longer timeseries of those atmospheric and land surface variables to infer the role of different ENSO types on the variability of these atmospheric and land variable thereby dust.

**Response:** Thank you so much for this instructive suggestion. It is true that the sample size is a big question of this study. For the role of different ENSO types, we moved it to the discussion section, where we discussed the possible influence of the ENSO type on the Atlantic and Indian ocean SST pattern that impact the ENSO-DUCMASS relationship. We also emphasized that the statistical results acquired from the insufficient number of samples could also be explained by the random events, and longer time series with valid samples (i.e., CT/EM ENSO and PDO years) are needed to further validate the influence of ENSO types on the ENSO–DUCMASS relationship in the future. Alternatively, using numerical model to simulate the teleconnection pattern of ENSO over South Asia under different types of ENSO is also favorable (lines 464-474). We are so sorry that we cannot analyze the role of different ENSO types on the variability of those atmospheric and land variable due to the limited dataset thereby limited sample size, nevertheless, we will repeat the same analysis to verify those conclusions in the future with more valid samples.

Comment 14: Finally, there are a lot of grammar errors and typos. Please check.

**Response:** Thank you so much for pointing out this to help us to improve our work. We are so sorry for making those mistakes. We have checked the whole text carefully and revised the tense errors, inappropriate words and typos. Thank you for your kind suggestions.

**Thank you again for your careful reading of our manuscript. We hope that the changes having been made to the manuscript meet to your satisfaction.**

**References:**

Babu SS, Manoj MR, Moorthy KK, et al. Trends in aerosol optical depth over Indian region: Potential causes and impact indicators[J/OL]. Journal of Geophysical Research Atmospheres, 2013, 118(20): 11,794-11,806. DOI:10.1002/2013JD020507.

Buchard V, Randles CA, Da Silva AM, et al. The MERRA-2 aerosol reanalysis, 1980 onward. Part II: Evaluation and case studies[J/OL]. Journal of Climate, 2017, 30(17): 6851-6872. DOI:10.1175/JCLI-D-16-0613.1.

Cherchi A, Navarra A. Influence of ENSO and of the Indian Ocean Dipole on the Indian summer monsoon variability[J/OL]. Climate Dynamics, 2013, 41(1): 81-103. DOI:10.1007/s00382-012-1602-y.

Clark CO, Cole JE, Webster PJ. Indian Ocean SST and Indian summer rainfall: Predictive relationships and their decadal variability[J/OL]. Journal of Climate, 2000, 13(14): 2503-2519. DOI:10.1175/1520-0442(2000)013<2503:IOSAIS>2.0.CO;2.

Kumar KK, Rajagopalan B, Cane MA. On the weakening relationship between the indian monsoon and ENSO[J/OL]. Science, 1999, 284(5423): 2156-2159. DOI:10.1126/science.284.5423.2156.

Randles CA, Sliva AM Da, Buchard V, et al. The MERRA-2 Aerosol Reanalysis, 1980 Onward. Part I: System Description and Data Assimilation Evaluation[J/OL]. Journal of Climate, 2017, 30(17): 6823-6850. DOI:10.1175/JCLI-D-16-0609.1.

Yang X, Huang P. Restored relationship between ENSO and Indian summer monsoon rainfall around 1999/2000[J/OL]. The Innovation, 2021, 2(2): 100102. DOI:10.1016/j.xinn.2021.100102.

---

## Author Comment (AC2)

Dear Professor:

Thank you for your kind comments for our manuscript entitled "**What caused the interdecadal shift of the ENSO impact on dust mass concentration over northwestern South Asia**", submitted to Atmospheric Chemistry and Physics. We appreciate your valuable comments and suggestions to improve it. We are sincerely grateful for giving us the opportunity to improve our work. With regard to your comments and suggestions, we wish to reply as follows:

Comment 1: Most of the arguments in this manuscript are mainly based on a series of sliding correlations (Figures 3, 4, 8, 11 & 12) and scatter diagrams (Figures 6, 7 & 9), without any analysis of dynamical processes associated with atmospheric circulation and climate elements. In fact, dust activities are mainly controlled by the local surface wind, precipitation or soil moisture (e.g., Goudie and Middleton, 1992). The teleconnection between the South Asian dust anomaly and the ENSO cycle is achieved through various mechanisms including the Walker Circulation variation (e.g., Huang et al., 2020). The existing analysis in this manuscript is not enough to establish the causal relationship between ENSO and the regional dust activities.

**Response:** Thank you so much for pointing out this to help us to improve our work. It is true that the analysis on influence mechanism is insufficient, which leads to great uncertainty in the relationship between ENSO and dust activities. According to your helpful suggestions, we compared the effect of ENSO and the considered influence factors (i.e., Atlantic SSTA index, Indian ocean SSTA index, and PDO) on the atmospheric circulation and climate elements, e.g., local precipitation, soil moisture, wind field, and velocity potential, to further verify the effects of those influence factors on the ENSO-dust relationship.

More specifically, for the influence mechanism of Atlantic SSTA index (ASGI), we compared the effect of North and South Atlantic SSTA with ENSO signal removed on the geopotential height at 850hPa/300hPa and zonal/meridional wind at 300hPa during the two study periods (Figs. 6-7), the results indicated that the response of atmospheric circulation on North and South Atlantic SSTA index modulated the ENSO-dust relationship. Besides, we compared the effect of ASGI and ENSO on local climate elements (soil moisture, precipitation, NDVI, wind field, and velocity potential) to prove that ASGI and ENSO exhibited the opposite effect on dust activities in the first period (Figs. 8-9).

As for the influence mechanism of Indian ocean SSTA index (TWISST), we compared the effect of TWISST and ENSO on local climate elements (soil moisture, precipitation, NDVI, wind field, and velocity potential) to prove the impact of TWISST on the ENSO-dust relationship (Figs. 12-14).

While for the influence mechanism of PDO, we compared the convection and its effect on dust activities between negative and positive phase of PDO as well as that between negative and positive phase of ENSO, the analogous effect of in-phase PDO and ENSO proved that PDO can strengthen the impact of ENSO on dust activities when it was in phase with ENSO (Fig. 16).

 This study discussed various factors influencing the interdecadal change of the ENSO-dust relationship, such as tropical Atlantic SST, Indian ocean SST, Eurasian continent temperature (or land-sea thermal contrast) and PDO. However, the role of some factors is unclear or unconvincing because the analysis is too superficial. Instead of dealing with so many factors in general, it is better to focus on one most important factor for in-depth discussion. So, I suggest deleting all parts related to factors insignificant for modulating the ENSO-dust relationship in the text and adding the relevant process analyses. For example, how does Atlantic SST affect the downstream dust? In particular, why can the spring SST control the summer dust? How can the large-scale SSTA affect the regional dust activities in northwestern South Asia far away from the SSTA region?

**Response:** Thank you so much for pointing out this to help us to improve our work. Our original intention was to explore the possible large-scale atmospheric factors that contributed to the interdecadal change of ENSO-dust relationship, next, we aimed to compare the effect of the three ocean SSTA pattern on the interdecadal change of this relationship since the significant influence of the three ocean SSTA pattern were found. Thus, we would like to keep these three subsections. However, we deleted the part related to factors insignificant for modulating the ENSO-dust relationship, i.e., the section discussing the effect of Eurasian continent and Indian Ocean thermal contrast. In addition, in order to avoid the random relationship, we added the analysis on the influence mechanism by examining their impacts on precipitation, soil moisture, NDVI, and wind, as that was stated in the response to the first comment.

Comment 3: In some figure captions (e.g., Figures 5 and 10), the months in which the SST are used should be clearly indicated. This is very important for how to explain the lag correlation between the SST and the dust. It seems that the March-May SST in the Atlantic Ocean while the September-May SST in the Indian Ocean has been used. However, the manuscript failed to give the physical mechanism through which the preceding SSTA affects the dust activity in the subsequent summer. Although several literatures have been cited to try to explain the possible connection between the two, for example, Atlantic SST affects the Indian monsoon (e.g., Rong et al., 2010; Kucharski and Joshi, 2017), most of these literatures only discuss the simultaneous correlation between the SST and the monsoon, rather than the lag correlation.

**Response:** Thank you so much for pointing out this to help us to perfect the results. We added months in which the SST are used in the figure captions, e.g., in Figures 3 and 5-7, the tropical SSTA was taken from spring (Mar.-May); in Figures 6-9 and 12-14, the geopotential height, soil moisture, precipitation, wind, and velocity potential were taken from the dust season (Jun.-Jul); in Figure 11, the tropical Indian SST was taken from the dust season (Jun.-Jul); in Figure 16, the velocity potential and wind were taken from the dust season (Jun.-Jul) and the PDO was taken from winter (Nov.-Jan.).

For the Atlantic, the SST averaged from Mar. to May was used in this study. We stated the reason as "According to Tokinaga et al. (2019), the Atlantic Niña pattern develops and is most sensitive to ENSO in spring, thus the SST averaged from Mar. to

May was used in this section" (lines 229-230). Previous studies also reported that the response of Atlantic to ENSO was strongest in spring, and this study also discussed the response of Atlantic SSTA to the two types of ENSO, thus we utilized the SST from Mar. to May.

As for the Indian ocean, the SST was the average of Jun.-Jul. that was synchronous with the dust season. We stated the reason as "Du et al. (2009) indicated that the North Indian Ocean warming displayed two peaks in Nov.–Dec.(-1) and Jun.–Aug.(0), with the second peak larger in magnitude. Cherchi and Navarra (2013) also pointed out that the connection between ISM and Indian ocean SST pattern was mostly confined in summer and autumn. Besides, compared to Atlantic, the Indian ocean is closer to the South Asian dust source, thus it takes less time to transmit the signal (partially through wave train propagation) from the Indian ocean to the dust source than that from the Atlantic. Given all of that, the Indian ocean SST used in this study was the summer average that was concurrent with the dust season (Jun.–Jul.)" (lines 308-315).

Overall, we only analyzed the impact of Atlantic, Indian ocean and Pacific SSTA in the season with the strongest signals, and failed to analyze the effect of those factors in various seasons. Besides, the physical mechanism of the lag effect was not discussed in detail and we just verified their impact on dust by examining their effects on some atmospheric circulation and climate elements. However, the mechanism of the lag effect was not in the scope of this study. We will consider it in the future work since it is a big question of this study.

Comment 4: In Line 132 of the text, it is mentioned that "the DUSMASS used in this study is averaged from June to July and May is neglected to weaken the disturbance of seasonal climatological differences". Actually, it is very important to understand the difference of ENSO impact on the pre-monsoon and monsoon season dust activities. This should be moderately discussed in the manuscript.

**Response:** Thank you so much for this creative suggestion. It is true that there is difference of ENSO impact on the pre-monsoon and monsoon season dust activities. According to your suggestion, we analyzed the interdecadal change of the correlation between ENSO and pre-monsoon dust activities, to find that the correlations were all insignificant during each period without significant interdecadal change. Thus we added the following statement: "The dust activities analyzed in this study were from the dust season, i.e., Jun.–Jul., which were part of monsoon season (Jun.–Sep.) (Babu et al., 2013), however, the dust activities during pre-monsoon season (Mar.–May or Apr.–May) were also a hot topic (Babu et al., 2013; Lakshmi et al., 2017, 2019). Therefore, we analyzed the interdecadal change of the ENSO impact on DUCMASS during dust season (Jun.–Jul.) and pre-monsoon (Mar.–May or Apr.–May) separately, to find that the significant interdecadal change occurred only when the DUCMASS during dust season was considered. As for that during pre-monsoon season, there should be some other factors that influenced its interdecadal change, which will be discussed in the future study". Considering that there is no more detail about the ENSO impact on pre-monsoon season dust activities, we added this statement into Sect. 4.3 rather than in a separate subsection.

**Thank you again for your careful reading of our manuscript. We hope that the changes having been made to the manuscript meet to your satisfaction.**

**References:**

Babu SS, Manoj MR, Moorthy KK, et al. Trends in aerosol optical depth over Indian region: Potential causes and impact indicators[J/OL]. Journal of Geophysical Research Atmospheres, 2013, 118(20): 11,794-11,806. DOI:10.1002/2013JD020507.

Cherchi A, Navarra A. Influence of ENSO and of the Indian Ocean Dipole on the Indian summer monsoon variability[J/OL]. Climate Dynamics, 2013, 41(1): 81-103. DOI:10.1007/s00382-012-1602-y.

Du Y, Xie SP, Huang G, et al. Role of air-sea interaction in the long persistence of El Niño-induced north Indian Ocean warming[J/OL]. Journal of Climate, 2009, 22(8): 2023-2038. DOI:10.1175/2008JCLI2590.1.

Lakshmi NB, Babu SS, Nair VS. Recent Regime Shifts in Mineral Dust Trends over South Asia from Long-Term CALIPSO Observations[J/OL]. IEEE Transactions on Geoscience and Remote Sensing, 2019, 57(7): 4485-4489. DOI:10.1109/TGRS.2019.2891338.

Lakshmi NB, Nair VS, Suresh Babu S. Vertical structure of aerosols and mineral dust over the Bay of Bengal from multisatellite observations[J/OL]. Journal of Geophysical Research: Atmospheres, 2017, 122(23): 12,845-12,861. DOI:10.1002/2017JD027643.

Tokinaga H, Richter I, Kosaka Y. ENSO Influence on the Atlantic Niño, Revisited: Multi-Year versus Single-Year ENSO Events[J]. Journal of Climate, 2019, 32(14): 4585-4600.

---

## Author Response (AR2)

**To reviewer #1**

**Dear Professor:**

Thank you for your kind comments for our manuscript entitled "What caused the interdecadal shift of the ENSO impact on dust mass concentration over northwestern South Asia", submitted to Atmospheric Chemistry and Physics. We appreciate your valuable comments and suggestions to improve it. We are sincerely grateful for giving us the opportunity to improve our work. With regard to your comments and suggestions, we wish to reply as follows:

1. While it's ok to analyze dust-PM2.5, the choice of analyzing dust-PM2.5 (rather than total DUCMASS or DUCMASS10) needs to be justified, since dust is typically regarded as a coarse-mode aerosol and usually have larger impacts on PM10 than PM2.5.

**Response:** Thank you so much for proposing this. It is true that dust is typically regarded as coarse mode aerosols that are closer to PM10 than PM2.5. Actually, at first, we used the DUCMASS dataset (which is also acquired from MERRA2), and later we compared the time series of DUCMASS and DUCMASS2.5, to find similar variation trend and the association of Niño index with DUCMASS and DUCMASS2.5 showed the same change pattern. Thus, we adopted the DUCMASS2.5 dataset at last, considering that PM2.5 is more widely used. According to your suggestions, we added the corresponding description in the dataset section, i.e., "The time series of DUCMASS25 dataset was compared with that of DUCMASS dataset, to find that the time series of DUCMASS25 and its association with Niño index showed the same change pattern with that of DUCMASS. Only the results acquired by DUCMASS25 were presented in this study". (Page 6 lines 143-146)

2. The caption of Figure 18 needs to be expanded. At least the authors need to point out that this is a Taylor diagram showing correlation and standard deviation of each data set relative to AERONET. The sample size should also be denoted on the corresponding panel of this figure. I would personally not call MERRA2 DUCMASS here as "high precision" given the moderate correlations.

**Response:** Thank you so much for pointing out this to help us to improve our work. We have changed the caption of Fig. 18 into "Normalized Taylor diagrams showing (a) difference between dust variables acquired from MISR, AERONET, MERRA2 datasets and that acquired from PM2.5 dataset, and (b-d) difference between dust variables from MISR, PM2.5, MERRA2 datasets and that acquired from AERONET dataset. The normalized standard deviation is on the radial axis (black contours); correlation coefficient is on the angular axis (blue contours); and green dashed lines indicate centered RMSE. N indicates the sample size". We also added sample size (n) in each subfigure. (Page 28 lines 541-546)

It is true that using "high precision" here is not appropriate. We changed the original "the DUCMASS used in this study were with high precision" into "the DUCMASS used in this study were relatively reliable for researches about dust". (Page 29 line 559)

**3. Please indicate statistical significance in Figures 3 and 5.**

**Response:** Thank you for pointing out this. We are so sorry for this negligence. We added significance test in Figs. 3, 5, and 11.

Thank you again for your careful reading of our manuscript. We hope that the changes having been made to the manuscript meet to your satisfaction.

**To reviewer #3**

**Dear Professor:**

Thank you for your kind comments for our manuscript entitled "What caused the interdecadal shift of the ENSO impact on dust mass concentration over northwestern South Asia", submitted to Atmospheric Chemistry and Physics. We appreciate your valuable comments and suggestions to improve it. We are sincerely grateful for giving us the opportunity to improve our work. With regard to your comments and suggestions, we wish to reply as follows:

**Major concerns:**

1. In the Introduction, the authors refer the human health and environmental problems. However, many studies have proved the climatic effects of dusts through different mechanisms (e.g., Miller and Tegen 1998; Mahowald et al., 2014). The authors should add the corresponding descriptions.

**Response:** Thank you so much for pointing out this to help us to improve our work. We added the corresponding descriptions as "Dust aerosols can also influence the earth's radiation budget balance and climate change through direct and indirect effects (Mahowald et al., 2014; Miller and Tegen, 1998). Dust aerosols can reflect incoming solar radiation and cool the surface, which is known as the direct effects (Mahowald et al., 2006; Tegen et al., 1996); they can also affect the cloud droplet size, cloud albedo and lifespan by forming into cloud condensation nuclei and ice nuclei, which is known as the indirect effects (Hansen et al., 1997)" in lines 35-40 of page 2.

2. As shown in Figs. 6-9, 10-14, and 16, these results are all based on regression of ENSO onto the climatic variables, the results are hard to understand for the readers. I strongly suggest to use the differences in climatic variables between El Niño (EN) and La Niña (LN) yeas to check the large-scale modes of climate variability (e.g., see the ref. Huang et al., 2020). These further results can help us to present the physical mechanisms or process research between ENSO and dust activity interactions.

**Response:** Thank you so much for pointing out this to help us to perfect the results. We agree with your opinion, in order to present the physical mechanisms of ENSO and dust activity interactions, it is better to use the differences in climatic variables between EN and LN years. However, this study intended to analyze the effect of other large-scale atmospheric factors (e.g., Atlantic and Indian ocean SSTA pattern) on the correlation between ENSO and dust activities in two periods (1982-1996 and 2000-2014). We should calculate the differences in climate variables between EN (ASGI+ and TWITSST+) and LN (ASGI+ and TWITSST-) years during the two periods separately, which make the sample size too small to get the robust results. Besides, we have learnt that regression method is also widely used in the influence mechanism analysis (Du et al., 2009; Han et al., 2016; England et al., 2014; Wu and Kirtman, 2004; Krishnan et al., 2013; Guo et al., 2019; Chattopadhyay et al., 2015), and using regression model can better show the change of the interaction between factors during the two periods, since the different signs of the correlation coefficient indicate the

contrary effect. Thus, we kept the results acquired by regression method.

At the same time, considering that your suggestions are helpful for us to clarify the physical mechanisms more clearly, we added the analysis about the differences in climate variables between EN and LN years as well as those between ASGI+/TWITSST+ and ASGI+/TWITSST- during the two periods separately. We got the similar results with the regression model, i.e., the positive regression coefficient appeared when the climatic variables and ENSO (or ASGI/TWITSST) were in phase, while negative regression coefficient appeared when the climatic variables and ENSO (or ASGI/TWITSST) were out of phase. Considering the similarity between the regression and composite difference results, we put the results demonstrated by the composite difference in the supplement file.

Could you please help us to judge whether it is ok to revise the corresponding parts like the abovementioned. Otherwise, we may modify it again according to your suggestions.

The modified parts are shown as follows:

**In section 3.2.1 (lines 303-315 of pages 14-15), we added:**

In addition, to further elaborate the physical mechanisms of the interaction between ENSO and dust activities, the composite differences of the abovementioned climatic variables between El Nino and La Nina years as well as that between positive ASGI (ASGI+) and negative ASGI (ASGI-) years were presented, as shown in Figs. S1–S2. Figure S1 showed that the SoilM averaged from June to July in El Niño years exhibited positive anomalies, while that in La Niña years exhibited the reversed anomalies. The differences of SoilM between ASGI+ and ASGI- during P1 were negative, which were contrary to that between El Niño and La Niña conditions, while the differences during P2 reversed compared with those in P1. Simultaneously, the differences of VP at 200hPa and 850hPa between ASGI+ and ASGI- also presented contrary change with those between El Niño and La Niña years during P1 and P2, as shown in Fig. S2. The mechanisms illustrated by the composite difference were analogous with the regression between dust activities and the climatic variables, both of which clarified the effect of ASGI on the relationship between ENSO and dust activities over the northwestern South Asian dust source.

Figure S1: June to July mean soil moisture (SoilM) averaged for the (a) El Niño years, (b) La Niña years, and (c) the difference between positive ASGI years (ASGI+) and negative ASGI (ASGI-) years during P1 (c) and P2 (d). The difference exceeding 90% confidence level was marked by black dots.